# β-amyloid and tau drive early Alzheimer's disease decline while glucose hypometabolism drives late decline

Tyler C. Hammond[1,2,11], Xin Xing[1,3,11], Chris Wang[3], David Ma[1,4], Kwangsik Nho[5], Paul K. Crane[6], Fanny Elahi[7], David A. Ziegler [7], Gongbo Liang [3], Qiang Cheng[8], Lucille M. Yanckello[1,9], Nathan Jacobs [3] & Ai-Ling Lin [1,2,9,10✉]

Clinical trials focusing on therapeutic candidates that modify β-amyloid (Aβ) have repeatedly failed to treat Alzheimer's disease (AD), suggesting that Aβ may not be the optimal target for treating AD. The evaluation of Aβ, tau, and neurodegenerative (A/T/N) biomarkers has been proposed for classifying AD. However, it remains unclear whether disturbances in each arm of the A/T/N framework contribute equally throughout the progression of AD. Here, using the random forest machine learning method to analyze participants in the Alzheimer's Disease Neuroimaging Initiative dataset, we show that A/T/N biomarkers show varying importance in predicting AD development, with elevated biomarkers of Aβ and tau better predicting early dementia status, and biomarkers of neurodegeneration, especially glucose hypometabolism, better predicting later dementia status. Our results suggest that AD treatments may also need to be disease stage-oriented with Aβ and tau as targets in early AD and glucose metabolism as a target in later AD.

[1] Sanders-Brown Center on Aging, University of Kentucky, 307 Sanders-Brown Building, 800 S. Limestone Street, Lexington, KY 40506-0230, USA. [2] Department of Neuroscience, University of Kentucky, Lexington, KY, USA. [3] Department of Computer Science, University of Kentucky, Lexington, KY, USA. [4] Department of Statistics, Harvard University, Boston, MA, USA. [5] Department of Radiology and Imaging Sciences, Indiana Alzheimer Disease Center, Center for Computational Biology and Bioinformatics, Indiana University School of Medicine, Indianapolis, IN, USA. [6] Department of Medicine, University of Washington, Seattle, WA, USA. [7] Department of Neurology, University of California, San Francisco, CA, USA. [8] Institute of Biomedical Informatics, University of Kentucky, Lexington, KY, USA. [9] Department of Pharmacology and Nutritional Sciences, University of Kentucky, Lexington, KY, USA. [10] F. Joseph Halcomb III, M.D. Department of Biomedical Engineering, University of Kentucky, Lexington, KY, USA. [11] These authors contributed equally: Tyler C. Hammond, Xin Xing. ✉email: ailing.lin@uky.edu

Alzheimer's disease (AD) is the most common form of dementia worldwide and is defined biologically as the pathologic deposition of folded β-amyloid (Aβ) plaques, and hyperphosphorylated neurofibrillary tau tangles in the brain leading to neurodegeneration[1–3]. Clinically, AD presents as a syndrome of progressive episodic memory and executive functioning problems across a cognitive continuum ranging through cognitively unimpaired (CU), mild cognitive impairment (MCI), and AD. While there are currently five drugs approved by the FDA to treat the symptoms of AD, there are no disease-modifying therapies that alter the course of the disease. Over the past few decades, the development of treatments for AD has been largely focused on compounds which aim to reduce Aβ plaques, either by directly targeting Aβ itself through antibodies or by targeting the enzymes that cleave amyloid precursor protein (APP) to produce it[4,5]. However, clinical trials of drugs targeting Aβ had a 99.6% failure rate between 2002 and 2012[6], and two more Aβ-focused drug trials failed in phase three in 2019[7]. This failure rate is among the highest of any disease area. The high failure rate for AD drug candidates focused on Aβ indicates that Aβ may not be the optimal therapeutic target to combat AD.

Careful analysis of AD biomarkers may give important insights into underlying AD pathogenesis and clues about appropriate AD treatments, since these biomarkers exist as proxies for AD neuropathologic changes. Furthermore, an understanding of how the biomarkers correlate with clinical symptoms of AD could inform clinicians making AD management decisithe ons to improve patient quality of life. The A/T/N biomarker framework promulgated by the National Institute on Aging-Alzheimer's Association was created to be an unbiased classification scheme for the three arms of biomarkers known to underlie AD pathology, namely neuropathological loads of Aβ (A) and tau (T), and neurodegeneration (N, including hypometabolism and brain atrophy)[8,9]. Indeed, some research groups have demonstrated that the distribution of tau tangles[10] and hypometabolism (due to low glucose uptake) are more strongly correlated with cognitive performance than Aβ[11]. Moreover, brain atrophy is also suggested to be highly correlated with AD progression[12]. However, it remains unclear whether disturbances in each arm of the A/T/N framework contribute equally to the progression of AD symptoms or if these factors instead have varying impacts at different stages of AD progression. Understanding this stage-dependent nature of the biomarkers could lead to important clues in preventing and treating AD.

In order to determine the nature of the association of AD biomarkers with the progression of AD symptoms, in this study, we assessed the statistical importance of each arm of the A/T/N framework in predicting three progressive clinical statuses of cognitive performance: cognitively unimpaired (CU), late mild cognitive impairment (LMCI), and AD[8]. To do so, we used data from the Alzheimer's Disease Neuroimaging Initiatives (ADNI) database, relating to four biomarkers: Aβ (assessed from [18]Florbetapir-positron emission tomography (PET)), phosphorylated tau (pTau181, assessed from cerebrospinal fluid), glucose uptake (assessed from [18]fluorodeoxyglucose (FDG)-PET), and volumetric measures (assessed from MRI). We used a random forest machine learning algorithm to rank the importance of each biomarker in predicting clinical dementia status. We chose the random forest machine learning method because it not only has the ability to fit models with high prediction accuracy due to its use of multiple decision trees that combine to yield a consensus prediction, but also is very interpretable due to its ranking capability of the relative importance of predictors used in the classification (AD biomarkers in our case). We also analyzed the relationship between A/T/N biomarkers and memory composite and executive functioning composite scores in order to assess more directly their association

with cognitive performance. We show that A/T/N biomarkers have differing contributions in predicting clinical dementia status based on the stage of cognitive impairment, with Aβ and pTau having higher contribution in predicting early cognitive impairment (LMCI vs. CU) and glucose uptake having higher contribution in predicting later cognitive impairment (AD vs. LMCI and AD vs. CU). Our findings could help real-world patient populations by informing clinicians to make AD management decisions according to disease stage based on the expression of the relevant A/T/N biomarkers, and by informing drug development teams to design treatments to target the pathophysiology underlying the expression of the biomarkers at the appropriate stage of disease progression.

## Results

**Participant characterizations and data selection**. Participant data was extracted from the ADNI database for inclusion in the analysis. Participants were required to have baseline Aβ imaging biomarkers (from [18]Florbetapir PET), glucose uptake imaging biomarkers (from [18]FDG PET), brain volume imaging biomarkers (from T1-weighted structural MRI), and cognitive testing to be included in the analysis. Participants with three or more missing values were excluded from the analysis. As tau imaging was not available for most participants in the ADNI database, we used a phosphorylated tau biomarker (pTau) from cerebrospinal fluid (CSF) as a measure of tau levels. These criteria yielded a final sample of 405 participants clinically diagnosed as being either cognitively unimpaired (CU; $n = 148$) or with LMCI; ($n = 147$) or Alzheimer's disease (AD; $n = 110$) (Table 1).

The three study groups were balanced in terms of gender, race, and ethnicity, but not age or education, across clinical status, with the AD group being significantly older than LMCI subjects and less educated than CU and LMCI subjects; accordingly, we adjusted the features for age before applying them to the random forest model since age is known to affect brain volumetric measures. Notably, the groups differed in terms of the expression of the ε4 allele of apolipoprotein E (APOE ε4), the largest genetic risk factor for Alzheimer's disease[13], and cognitive testing scores, with the AD group being significantly more likely to carry APOE ε4 and to have lower cognitive testing scores than CU and LMCI subjects. The cognitive tests completed included the mini-mental state examination, clinical dementia rating sum of boxes, Alzheimer's disease assessment scale-cognitive subscale (ADAS-cog 13), composite memory score (ADNI_MEM), and composite executive functioning score (ADNI_EF). The biomarkers were further stratified into 16 features classified according to the A/T/N framework, comprising Aβ measures from six brain regions (frontal lobe, cingulate gyrus, parietal lobe, temporal lobe, precuneus, and hippocampus), glucose uptake (FDG) data from three brain regions (angular gyrus, temporal lobe, and posterior cingulum), volumetric measures from six regions (ventricles, whole brain, entorhinal cortex, hippocampus, gray matter, and white matter), and pTau levels from the CSF (Table 2). We show the correlation of the 16 features with each other using a heatmap (Supplementary Fig. 1). It shows that the Aβ measures were highly correlated with each other, as were the FDG measures, and the volumetric measures, while Aβ and pTau were negatively correlated with FDG and volumetric measures.

**Relative importance of AD biomarkers in early and late AD.** We first sought to determine the relative importance of each biomarker feature in predicting clinical dementia status in three participant group pairings: CU vs. LMCI, LMCI vs. AD, and CU vs. AD. Table 3 shows the descending order ranking of the relative importance of the 16 features in predicting clinical

**Table 1 Demographic and cognitive data for the cross-sectional study population.**

|  | CU | LMCI | AD | $\chi^2$-approx | $\varepsilon^2$ | P-value |
|---|---|---|---|---|---|---|
| **Subject characteristics** | | | | | | |
| n | 148 | 147 | 110 | | | |
| Age (years) | 73.43 ± 6.29 | 71.98 ± 7.42 | 74.46 ± 8.39 | 7.207 | 0.0178 | 0.0272* |
| Gender (% Male) | 51% | 54% | 60% | 2.236 | 0.00554 | 0.3268 |
| Education (years) | 16.63 ± 2.53 | 16.70 ± 2.45 | 15.61 ± 2.55 | 13.395 | 0.0332 | 0.0012* |
| *APOE ε4* carriers (%) | 27% | 57% | 69% | 53.653 | 0.133 | <0.0001* |
| Ethnicity (% Hispanic) | 5.4% | 1.4% | 3.6% | 3.673 | 0.00909 | 0.1594 |
| Race (% White) | 89% | 95% | 92% | 2.799 | 0.00693 | 0.2467 |
| (% Black) | 7% | 3% | 4% | | | |
| (% Asian) | 2% | 1% | 4% | | | |
| **Cognitive data** | | | | | | |
| MMSE | 29.06 ± 1.14 | 27.61 ± 1.82 | 23.14 ± 2.03 | 246.414 | 0.61 | <0.0001* |
| CDRSB | 0.03 ± 0.13 | 1.71 ± 1.00 | 4.60 ± 1.61 | 351.755 | 0.871 | <0.0001* |
| ADAS-cog 13 | 9.08 ± 4.58 | 18.57 ± 7.08 | 30.16 ± 9.70 | 239.827 | 0.594 | <0.0001* |
| ADNI_MEM | 1.06 ± 0.63 | −0.03 ± 0.66 | −0.89 ± 0.54 | 266.260 | 0.63 | <0.0001* |
| ADNI_EF | 0.94 ± 0.81 | 0.16 ± 0.85 | −0.83 ± 0.93 | 161.477 | 0.388 | <0.0001* |

Values are displayed as the mean ± SD. The $\chi^2$-approx test statistic is calculated from a Kruskal–Wallis test comparing the groups CU, LMCI, and AD. $\varepsilon^2$ is the effect size calculated from a Kruskal–Wallis test. Asterisk (*) next to P-value indicates statistical significance. DF = 2 for all comparisons.
CU, cognitively unimpaired; LMCI, late mild cognitive impairment; AD, Alzheimer's disease; MMSE, mini-mental state examination; CDRSB, clinical dementia rating sum of boxes; ADAS-cog, Alzheimer's disease assessment scale-cognitive subscale; ADNI_MEM, composite memory score; ADNI_EF, composite executive functioning score.

**Table 2 Biomarkers used in the feature analysis.**

| Data source | Biomarker measure | Features | A/T/N classification |
|---|---|---|---|
| Positron emission tomography (PET) | Amyloid-beta (AV45; [18]Florbetapir) | 1. Aβ-Frontal<br>2. Aβ-Cingulate<br>3. Aβ-Parietal<br>4. Aβ-Temporal<br>5. Aβ-Precuneus<br>6. Aβ-Hippocampus | A |
| | Glucose uptake ([18]FDG) | 7. FDG-Angular<br>8. FDG-Temporal<br>9. FDG-CingulumPost | N |
| Magnetic resonance imaging (MRI) | Volumetric measures | 10. Ventricle volume<br>11. Whole brain volume (WBV)<br>12. Entorhinal cortex volume<br>13. Hippocampal volume<br>14. Gray matter volume (GMV)<br>15. White matter volume (WMV) | |
| Cerebrospinal fluid (CSF) | phosphor-Tau ([181]P) | 16. Phosphorylated tau (pTau) | T |

Amyloid-beta measures include Aβ from the frontal lobe (Aβ-Frontal), cingulate cortex (Aβ-Cingulate), parietal lobe (Aβ-Parietal), temporal lobe (Aβ -Temporal), precuneus (Aβ -Precuneus), and hippocampus (Aβ -Hippocampus).
Glucose uptake measures include FDG from the angular gyrus (FDG-Angular), temporal lobe (FDG-Temporal), and posterior cingulum (FDG CingulumPost).
[18]FDG, Fluorodeoxyglucose.

dementia status based on the random forest method, a machine-learning algorithm that utilizes multiple decision trees to classify and rank variables according to their accuracy in predicting outcomes. Notably, the top half (top 8) of the features made up the majority of the relative importance (69.1, 75.45, and 86.74%) for each cognitive state classification. In CU vs. LMCI, hippocampal volume ranked highest in relative prediction accuracy with a relative importance of 12.69%, followed by four Aβ features, pTau, FDG in the angular gyrus (FDG-Angular), and entorhinal cortex volume; thus, features from all three arms of the A/T/N framework were represented in the top eight features in the CU vs. LMCI comparison. In contrast, in LMCI vs. AD, neurodegeneration, i.e., the N component of the framework, dominated the top eight features, with all three FDG glucose uptake measurements (temporal lobe = 18.88% relative importance) and entorhinal cortex, hippocampal and ventricle volumes represented. In particular, the three FDG features were ranked as the top three contributors in the LMCI vs. AD comparison. Moreover, the contribution of FDG was weighted even higher in CU vs. AD, with FDG-Angular making up 23.78%, and FDG in the posterior cingulum (FDG-CingulumPost) making up 16.99% of the relative importance. Another N component, hippocampal volume, also had an increased relative importance in the CU vs. AD comparison relative to the other comparisons. The findings suggest that, overall, Aβ and pTau are important contributors to the progression from normal cognitive functioning to LMCI, but that neurodegeneration, especially glucose hypometabolism, emerges as a more important contributor when progressing from LMCI to AD. Glucose hypometabolism also serves as a prominent distinguishing feature between normal cognitive functioning and AD. We replicated our analysis using the SHapley Additive exPlanations (SHAP) technique and obtained a feature ranking analysis consistent with those from the random forest analysis (Supplementary Fig. 2).

**Table 3 Ranking of each biomarker feature importance to prediction of diagnosis classification from the random forest analysis.**

| | Rank | CU vs. LMCI | | LMCI vs. AD | | CU vs. AD | |
|---|---|---|---|---|---|---|---|
| | | Biomarker feature | Relative importance (%) | Biomarker feature | Relative importance (%) | Biomarker feature | Relative importance (%) |
| Top half | 1 | Hippocampus volume | 12.69 | FDG-temporal | 18.88 | FDG-angular | 23.78 |
| | 2 | Aβ-frontal | 11.51 | FDG-angular | 17.36 | FDG-CingulumPost | 16.99 |
| | 3 | Aβ-temporal | 8.57 | FDG-CingulumPost | 12.11 | Hippocampus volume | 12.93 |
| | 4 | FDG-angular | 8.32 | Hippocampus volume | 7.49 | FDG-temporal | 10.00 |
| | 5 | Entorhinal cortex volume | 7.88 | Aβ-precuneus | 5.14 | Aβ-temporal | 8.11 |
| | 6 | Aβ-precuneus | 7.78 | Aβ-temporal | 4.97 | Aβ-precuneus | 6.20 |
| | 7 | pTau | 7.60 | pTau | 4.82 | Entorhinal cortex volume | 4.81 |
| | 8 | Aβ-cingulate | 4.75 | Entorhinal cortex volume | 4.76 | pTau | 3.92 |
| | Subtotal | | 69.1 | | 75.45 | | 86.74 |
| Bottom half | 9 | Aβ-hippocampus | 4.48 | Aβ-parietal | 4.71 | Aβ-frontal | 3.81 |
| | 10 | Ventricles | 4.35 | Aβ-frontal | 3.98 | Aβ-parietal | 3.32 |
| | 11 | FDG-CingulumPost | 4.29 | Ventricles | 3.69 | Aβ-Hippocampus | 3.17 |
| | 12 | GMV | 4.20 | Aβ-Hippocampus | 3.14 | Aβ-Cingulate | 0.99 |
| | 13 | WMV | 3.56 | Aβ-Cingulate | 2.63 | Ventricles | 0.65 |
| | 14 | FDG-temporal | 3.48 | GMV | 2.59 | GMV | 0.60 |
| | 15 | WBV | 3.45 | WBV | 2.46 | WMV | 0.38 |
| | 16 | Aβ-parietal | 3.10 | WMV | 1.29 | WBV | 0.32 |
| | Subtotal | | 30.9 | | 24.55 | | 13.26 |
| Sum | | | 100 | | 100 | | 100 |

CU, cognitively unimpaired; LMCI, late mild cognitive impairment; AD, Alzheimer's disease; FDG, fluorodeoxyglucose; GMV, gray matter volume; WMV, white matter volume; WBV, whole brain volume.

**Table 4 Accuracy of all 16 features and of the top 8 features in predicting diagnosis for each participant group comparison.**

All 16 features

| | CU vs. LMCI | LMCI vs. AD | CU vs. AD |
|---|---|---|---|
| Accuracy (%) | 73.17 | 71.01 | 90.34 |
| $F_1$ score (%) | 73.09 | 70.84 | 90.32 |

Top 8 features

| | CU vs. LMCI | LMCI vs. AD | CU vs. AD |
|---|---|---|---|
| Accuracy (%) | 72.74 | 70.15 | 91.63 |
| $F_1$ score (%) | 72.59 | 70.02 | 91.59 |

**Accuracy of the top 8 features vs. all 16 features**. We next determined the prediction accuracy of all 16 features in classifying the three participant group pairs. For all 16 features, accuracies of 73.17%, 71.01%, and 90.34% were obtained for the CU vs. LMCI, LMCI vs. AD, and CU vs. AD comparisons, respectively (Table 4). To ensure that our cognitive status classification model was robust, the F1 score was also used to evaluate the precision and recall of the model. The results show that the 16 features were able to classify the three group pairs with high accuracy. Knowing that the top eight biomarker features have high relative importance in predicting cognitive status, we also explored whether the classification accuracy of the top eight features was comparable to that of all 16 features. Using the top eight features only, accuracies of 72.74%, 70.15%, and 91.63% were obtained for the CU vs. LMCI, LMCI vs. AD, and CU vs. AD comparisons, respectively, thus confirming that the accuracy of the top eight features was similar to that of all 16 features. Figure 1 depicts the comparison of the receiver operating characteristic curves with five-fold cross validation[14] between all 16 features and between the top 8 features. We found similar results in accuracy when using three-fold and ten-fold cross validations for

comparison (Supplementary Table 1). The ROCs show that all 16 features performed slightly better than the top eight features in distinguishing CU vs. LMCI and LMCI vs. AD. Precision recall (PR) curves verified similar levels of accuracy (Supplementary Fig. 3). These results suggest that there may be feature redundancy present when all 16 features are used to predict cognitive state: indeed, we found some of the features to be insignificant for cognitive state classification, especially those with the lowest ranking. Our findings suggest that the accuracy of the cognitive state prediction model does not depend strictly on the number of features used in the model, and that the top eight features may be sufficient to accurately classify the three clinical cognitive statuses.

**Correlation of AD biomarkers with cognitive performance**. To understand if the top eight features in classifying the three participant group pairs are associated with performance on memory and executive functioning tests, we performed a correlation analysis of each feature on a memory composite score[15] and on an executive functioning composite score[16]. These composite scores are validated, psychometrically sophisticated composite scores based on the ADNI battery of neuropsychological tests described above. Ranking the biomarker features based on their r correlation value, we found that the pattern of biomarker correlation with performance on memory and executive functioning tests across participant groups was similar to the pattern found in the feature ranking analysis. When comparing the CU and LMCI groups (Fig. 2a), memory performance was inversely correlated with Aβ biomarkers, especially Aβ in the temporal (Aβ-temporal), Aβ in the precuneus (Aβ-precuneus), and Aβ in the frontal lobe (Aβ-frontal). Hippocampal volume was also highly positively correlated and pTau was highly negatively correlated with memory when comparing CU and LMCI. However, when comparing LMCI and AD data (Fig. 2b), in all three brain areas assessed, glucose uptake (FDG) was the feature most highly positively correlated with memory, showing larger

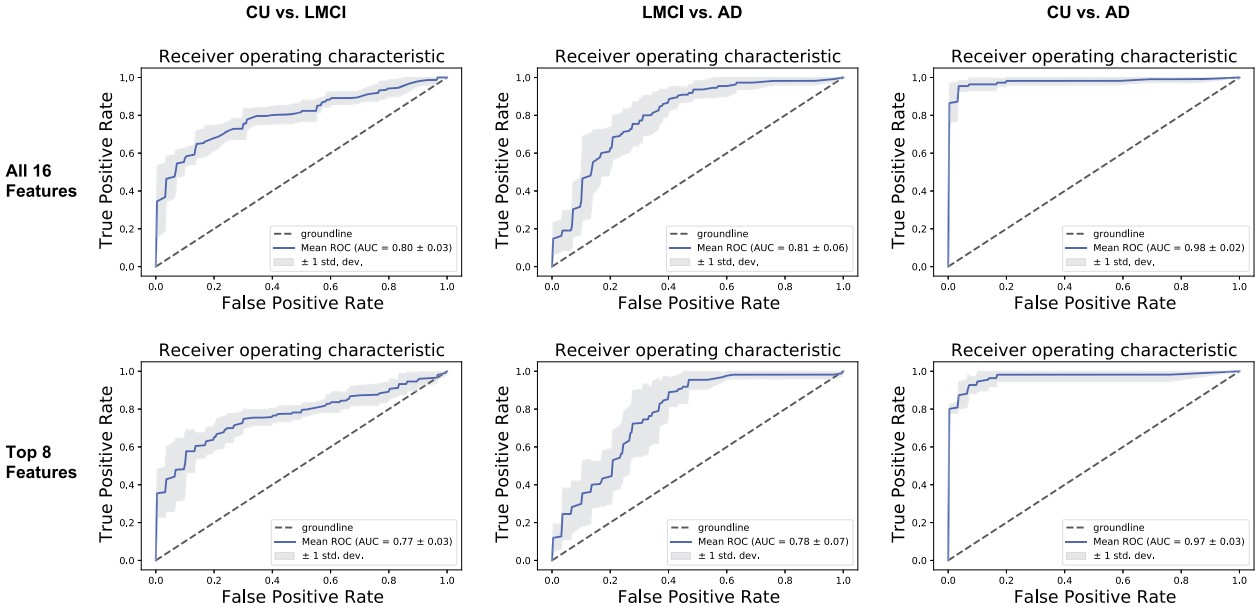

**Fig. 1 Receiver operating characteristic (ROC) curves depicting the accuracy of all 16 biomarker features (top) vs. the top 8 biomarker features (bottom).** Comparison of receiver operating characteristic[14] curves between all 16 biomarker features (top) and the top 8 biomarker features (bottom) from the three diagnosis participant group comparisons: cognitively unimpaired (CU) vs. late mild cognitive impairment (LMCI), LCMI vs. Alzheimer's disease (AD), and CU vs. AD. Groundline refers to a model that cannot predict better than random chance. The mean ROC is calculated from the average of the five ROC curves produced from the k-fold cross validation.

correlation coefficients (*r* values) than those in the CU vs. LMCI analysis. A similar correlation pattern was observed when comparing CU and AD data (Fig. 2c), with the correlation constants being even larger for the FDG measurements than in the LMCI and AD comparison. These results suggest that FDG biomarkers become increasingly predictive of memory performance as cognitive decline progresses from LMCI to AD. In particular, FDG-angular appears to be an especially important predictor of memory function, as it has the highest correlation coefficient of the three FDG biomarkers in these memory correlation analyses. A similar pattern to that observed in the memory performance analyses emerged when correlating executive functioning with the top eight biomarkers with FDG biomarkers becoming increasingly predictive of executive functioning as cognitive decline progresses (Fig. 3). Notably, however, pTau and Aβ-Precuneus were more highly correlated with memory than executive functioning in the CU vs. LMCI group (Figs. 2a and 3a). Interestingly, out of all the features, pTau showed the smallest correlation with executive functioning in each group (Fig. 3a–c).

**Biomarker quantification for predicting LMCI and AD.** Having shown through ranking and correlation that the top eight features from each participant group may be used as effective biomarkers to predict disease progression from CU to LMCI and AD, we next sought to assess the average values for each biomarker feature in each diagnosis group that can be used for the clinical diagnosis of these three cognitive statuses. Table 5 summarizes the values of each of the top eight features that can be used to distinguish CU, LMCI, and AD.

## Discussion

We demonstrated three novel aspects in this study. First, we employed AD biomarkers from all arms of the newly developed A/T/N framework in a random forest machine learning analysis powerful enough to accurately predict an AD diagnosis of CU, LMCI, or AD and to rank biomarkers in order of their

importance in the prediction. Second, we showed that biomarkers from the A/T/N framework have differing importance in predicting clinical dementia status across the disease progression, with Aβ and pTau having higher importance in predicting early cognitive impairment (CU vs. LMCI) and glucose uptake having higher importance in predicting later cognitive impairment (LMCI vs. AD and CU vs. AD) (Fig. 4). Our findings suggest that Aβ and pTau accumulation contribute to the cognitive decline that leads to LMCI, but may not be sufficient to lead to clinical AD. Instead, neurodegeneration, especially in the form of glucose hypometabolism, appears to be crucial for exacerbating cognitive decline and furthering its progression to clinical AD. Additionally, we found that Aβ and pTau are more strongly correlated with cognitive performance in LMCI, while glucose hypometabolism is more strongly correlated with cognitive performance in AD, with FDG biomarkers becoming increasingly predictive of memory and executive functioning as cognitive decline progresses. While others have previously documented the temporal ordering of biomarkers preceding clinical symptomatology of Alzheimer's disease[17], the real strength of our analysis is in creating algorithms for computational analyses that are consistent with available clinical and imaging data from data that has been collected over many years. The challenge moving forward will be to translate these algorithms into usable tools to that can assess the capacity of patients in a clinically-friendly manner. Finally, we demonstrated that the top eight features used in classifying the three participant group pairs were just as accurate in predicting clinical dementia status as all 16 features combined. The top eight biomarker features that can be used to distinguish between stages of cognitive impairment, which may prove useful for the future prediction and diagnosis of LMCI and AD.

Machine learning techniques have previously been used to predict cognitive status in AD using several separate biomarkers, including those measured by FDG-PET[18,19], structural MRI[18,20–22], amyloid-PET[20,23,24], and CSF-phosphorylated tau[21,22]. However, this is the first study to our knowledge to combine biomarkers from all arms of the A/T/N framework

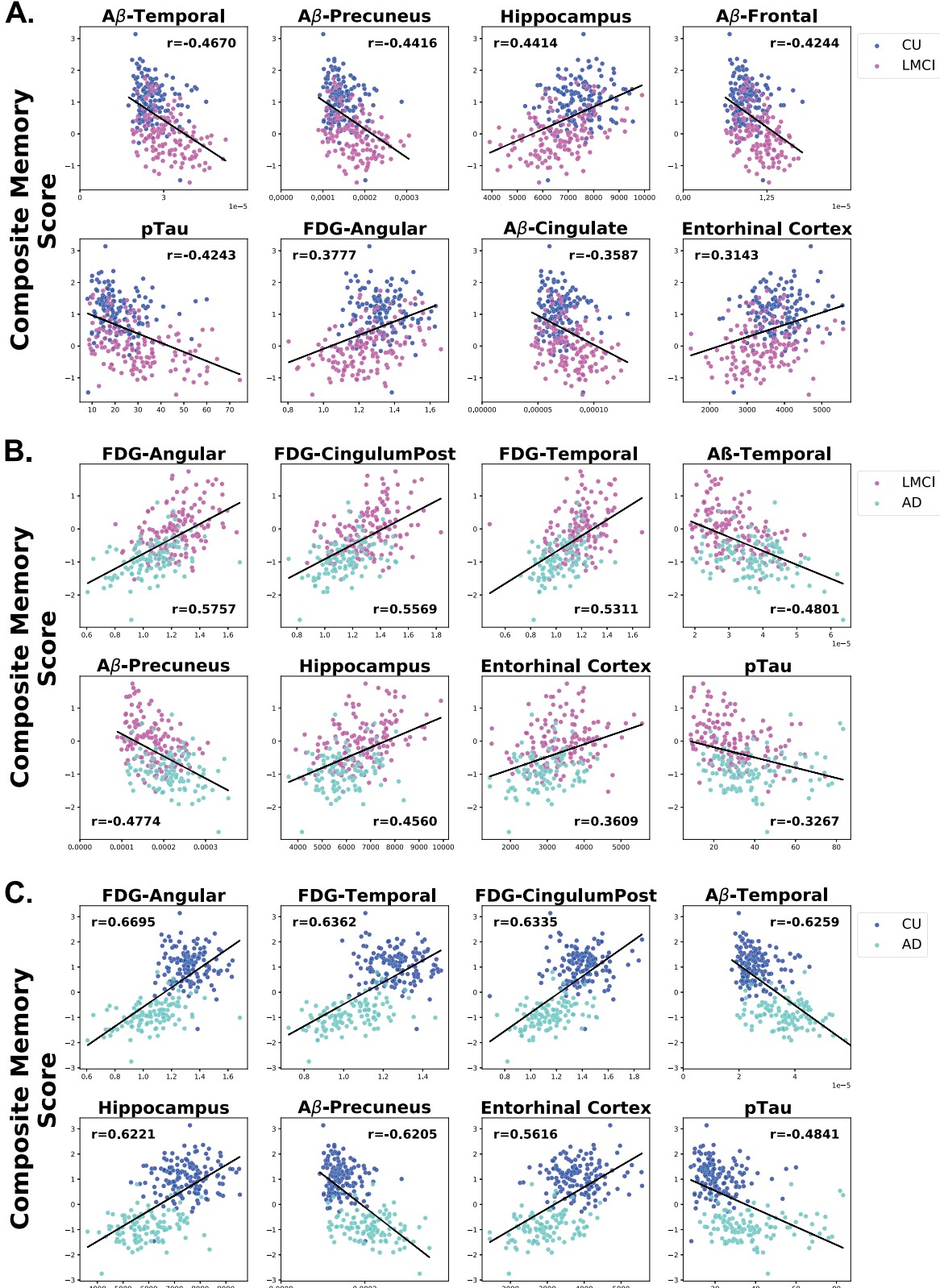

**Fig. 2 Correlation of top eight AD biomarkers with composite memory scores.** Scatter plots showing the correlations of the top eight features with performance on composite memory tests in each pairwise analysis among the cognitive statuses. (**a**) CU vs. LMCI. (**b**) LMCI vs. AD. (**c**) CU vs. AD. The order of the scatter plots in each panel is according to the rank of the *r* correlation value when compared to composite memory score. The *x*-axis refers to the indicated biomarker score and the *y*-axis refers to the composite memory score. Each dot refers to the indicated biomarker score and composite memory score of a single participant.

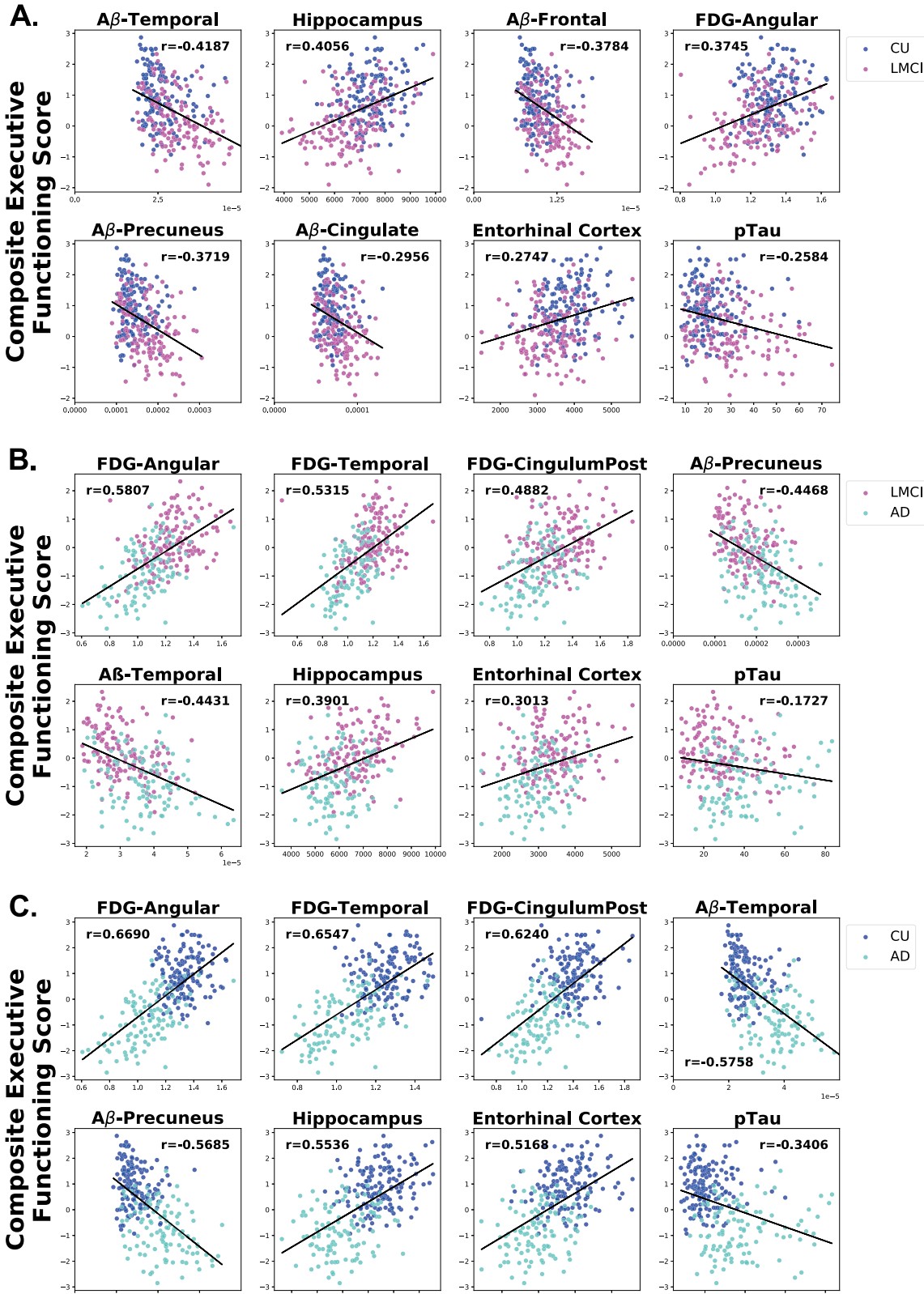

**Fig. 3 Correlation of top eight AD biomarkers with executive functioning scores.** Scatter plots showing the correlations of the top eight features with performance on composite executive functioning tests in each pairwise analysis among the cognitive statuses (**a**) CU vs. LMCI. (**b**) LMCI vs. AD. (**c**) CU vs. AD. The order of the scatter plots in each panel is according to the rank of the *r* correlation value when compared to composite executive functioning score. The *x*-axis refers to the indicated biomarker score and the *y*-axis refers to the composite memory score. Each dot refers to the indicated biomarker score and composite executive functioning score of a single participant.

**Table 5 Average values of the top eight biomarker features for each diagnosis group that can be used to predict cognitive status.**

| Features | CU | LMCI | AD | A/T/N arm |
|---|---|---|---|---|
| Aβ-Precuneus (SUV cm$^{-3}$) | 0.0715 ± 0.0154 | 0.0873 ± 0.0231 <br> $Z = 5.79$, $P < 0.0001$** $r = 0.34$ | 0.107 ± 0.0272 <br> $Z = 9.89$, $P < 0.0001$‡‡ $r = 0.62$ | A |
| Aβ-Frontal (SUV cm$^{-3}$) | 0.00949 ± 0.00187 | 0.0114 ± 0.00260 <br> $Z = 6.75$, $P < 0.0001$** $r = 0.39$ | 0.0129 ± 0.00286 <br> $Z = 9.07$, $P < 0.0001$‡‡ $r = 0.56$ | |
| Aβ-Cingulate (SUV cm$^{-3}$) | 0.0692 ± 0.00111 | 0.0787 ± 0.00181 <br> $Z = 4.87$, $P < 0.0001$** $r = 0.28$ | 0.0892 ± 0.00192 <br> $Z = 8.03$, $P < 0.0001$‡‡ $r = 0.50$ | |
| Aβ-Temporal (SUV cm$^{-3}$) | 0.0259 ± 0.00523 | 0.0302 ± 0.00678 <br> $Z = 6.76$, $P < 0.0001$** $r = 0.39$ | 0.0331 ± 0.00672 <br> $Z = 9.92$, $P < 0.0001$‡‡ $r = 0.62$ | |
| pTau (pg ml$^{-1}$) | 21.50 ± 8.87 | 29.70 ± 14.01 <br> $Z = 5.62$, $P < 0.0001$** $r = 0.33$ | 38.50 ± 16.52 <br> $Z = 9.56$, $P < 0.0001$‡‡ $r = 0.60$ | T |
| FDG-Angular (SUV cm$^{-3}$) | 1.21 ± 0.104 | 1.13 ± 0.149 <br> $Z = 5.31$, $P < 0.0001$** $r = 0.31$ | 0.956 ± 0.159 <br> $Z = 11.46$, $P < 0.0001$‡‡ $r = 0.71$ | N |
| FDG-CingulumPost (SUV cm$^{-3}$) | 3.03 ± 0.324 | 2.84 ± 0.391 <br> $Z = 4.72$, $P < 0.0001$** $r = 0.27$ | 2.47 ± 0.343 <br> $Z = 10.69$, $P < 0.0001$‡‡ $r = 0.67$ | |
| FDG-Temporal (SUV cm$^{-3}$) | 8.24 ± 0.706 | 7.78 ± 0.983 <br> $Z = 4.54$, $P < 0.0001$** $r = 0.26$ | 6.73 ± 0.924 <br> $Z = 10.82$, $P < 0.0001$‡‡ $r = 0.67$ | |
| Hippocampus volume (cm$^3$) | 7.49 ± 0.827 | 6.67 ± 1.11 <br> $Z = 6.62$, $P < 0.0001$** $r = 0.39$ | 5.91 ± 0.923 <br> $Z=10.59$, $p < 0.0001$‡‡ $r=0.66$ | |
| Entorhinal cortex volume (cm$^3$) | 3.85 ± 0.587 | 3.39 ± 0.710 <br> $Z = 5.51$, $P < 0.0001$** $r = 0.32$ | 2.92 ± 0.622 <br> $Z = 9.40$, $P < 0.0001$‡‡ $r = 0.59$ | |

Values are displayed as the mean ± SD.
**$P < 0.0001$ calculated with a Wilcoxon rank-sum test comparing CU vs. LMCI.
‡‡$P < 0.0001$ calculated with a Wilcoxon rank-sum test comparing CU vs. AD.
SUV is the standard uptake value, $Z$ is the $Z$-score test statistic for Wilcoxon rank-sum test, $r$ is the effect size for Wilcoxon rank-sum test.

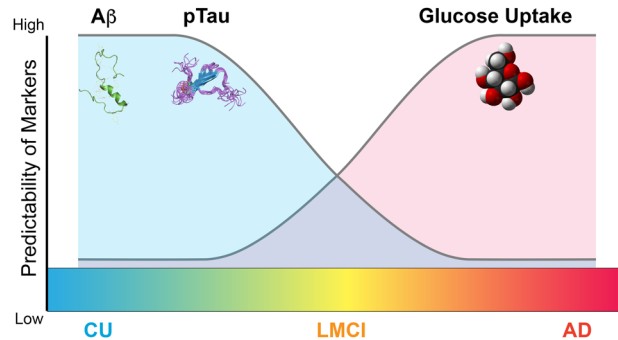

**Fig. 4 Relative importance of biomarkers predicting AD clinical diagnosis.** Diagram depicting the relative importance of biomarkers in predicting AD clinical diagnosis (predictability). In early AD, Aβ and pTau deposition in the brain have higher relative importance in predicting AD clinical diagnosis. In late disease low glucose uptake in the brain has higher relative importance in predicting AD clinical diagnosis.

into one integrated analysis using a machine learning method capable of classifying clinical dementia status and ranking the biomarker features according to their relative importance in the prediction model. Consistent with our findings, a previous study showed that Aβ was more highly associated with cognitive decline in cognitively normal participants, while glucose hypometabolism was more closely linked with cognitive decline in moderate and later stages of the disease (LMCI/AD)[25]. Additionally, another study showed that FDG-PET, which assesses glucose uptake, is more highly correlated to cognitive ability than Aβ levels in patients with MCI and AD[11]. These studies, in conjunction with our findings strongly support the argument that cognitive decline in AD is initially propagated by Aβ and tau aggregation but is further exacerbated by glucose hypometabolism as cognitive decline progresses. Our findings could better inform clinical AD management decisions and may

shift the targets of therapies to treat and prevent AD in future drug development.

We found that Aβ and pTau accumulation are more highly correlated with cognitive test scores in the CU vs. LMCI comparison than other biomarker features. In particular, Aβ deposition in the temporal cortex, precuneus, and frontal cortex, as well as increased hippocampal volume, appear to be the most important features in predicting memory and executive functioning performance in early stage disease. Indeed, these areas play a central role in a wide spectrum of highly integrated tasks that are noticeably disturbed in patients with MCI. For example, the temporal cortex is involved in memory, auditory cognition and semantics[26]; the precuneus is involved in visuo-spatial image processing and episodic memory retrieval[27]; the frontal lobe is involved in executive function, attention, memory, and language[28]; and the hippocampus is important for declarative memory[29]. Additionally, we found that increased levels of pTau were associated with memory performance but not executive functioning in LMCI, which is consistent with previous findings[30]. We note that some groups have found a high correlation between tau levels and cognitive decline across the entire AD spectrum[31,32]; even so, our results align with those of Mielke et al., who found a significant association between tau and cognitive performance in MCI, but a nonsignificant association between these factors in AD[33]. We also noticed that the atrophy of the hippocampus and entorhinal cortex (measures of neurodegeneration) were highly correlated in the CU vs. LMCI comparison of cognitive test scores; in addition to Aβ and pTau burden, brain atrophy in these two regions may thus substantially contribute to progression from CU to LMCI status, as other groups have reported[34].

We observed that impaired glucose uptake is most highly correlated with cognitive test scores in LMCI vs. AD and CU vs. AD groups. In particular, we found glucose uptake in the angular gyrus (FDG-angular) to be the most important feature for predicting memory and executive functioning performance in later

stages of AD, which is consistent with other groups who have found reduced glucose uptake in the angular gyrus in later cognitive decline[35]. This area is involved in semantic processing, word comprehension, number processing, memory retrieval, attention, spatial and social cognition, and reasoning[36], all of which are known to decline later in disease progression. Sustained deficits in glucose uptake in key brain areas dramatically impair cognitive functions by reducing proper support of neuronal activity and functional processes[37–39], and it is therefore unsurprising that we found that impaired glucose uptake is highly correlated with advancing cognitive decline.

Notably, the individuals with AD that were included in the current study were older and less educated than individuals in other groups, and a higher percentage of AD patients carried the APOE ε4 allele, the largest genetic risk factor for AD, than LMCI and CU patients. Interestingly, all three of these factors are linked to metabolic function[40–44]. A widely accepted cause of the functional losses that accompany aging is decreased brain metabolic function[45,46]. Indeed, mitochondrial function declines with age in the brain and, thus, neural ATP production decreases, which has been proposed to be a major factor in the aging-associated loss of brain function[40,43,46]. Moreover, a recent study demonstrated that regional brain metabolism and functional connectivity as measured by fMRI differed with years of education[41]: relative to less educated participants, highly educated participants had higher glucose metabolism in the ventral areas of the cerebrum, which are mainly involved in memory, language, and neurogenesis, and functional connectivity experiments illustrated that the brains of the highly educated individuals were overall more efficient and resilient to aging[41]. The APOE gene plays a role in cholesterol and Aβ homeostasis[39], and the APOE ε4 allele is the strongest genetic risk factor for AD. Two recent studies showed that disturbances in cholesterol metabolism, such as alterations in bile acid metabolism, are highly associated with AD[47,48]. Notably, the bile acid composition signatures were much more highly associated with brain hypometabolism and atrophy (i.e., the "N" component of the A/T/N framework) than with Aβ and tau. Moreover, cross-sectional FDG-PET studies found that cognitively unimpaired carriers of the APOE ε4 allele have abnormally low glucose uptake in the same brain regions that show hypometabolism in AD patients. Indeed, these metabolic abnormalities were observed in late-middle-aged (40–60 years of age) and young (20–39 years of age) APOE ε4 carriers, who had intact memory and were free of Aβ or tau pathology[49–53]. These neuroimaging results suggest that APOE ε4 carriers develop functional brain abnormalities several decades before the possible onset of dementia, and the results are in line with our finding that a high percentage of those with clinical AD were APOE ε4 carriers.

There are many plausible reasons to explain why we found glucose hypometabolism to be an important biomarker in predicting progressive cognitive decline in clinical AD. For example, impairments in brain glucose metabolism are associated with insulin resistance, which, in turn, exacerbates Aβ deposition[39,54]. Indeed, AD is characterized by impaired brain insulin signaling[55]. In line with this finding, type 2 diabetes mellitus, hyperlipidemia, obesity, and other metabolic diseases increase the risk of developing AD[12,39]. Indeed the metabolic abnormalities present in AD are often likened to a form of diabetes of the brain[56]. The preservation of normal brain glucose metabolism is, thus, highly associated with cognitive resilience. A recent study showed that FDG-PET uptake in the bilateral anterior cingulate cortex and anterior temporal pole was positively associated with global cognition in cognitively unimpaired individuals over 80 years of age, despite the fact that they were Aβ-positive and APOE ε4-positive[44]. The results also suggest that normal cognitive performance can be preserved even in the presence of Aβ and APOE ε4 in 80+ year-old individuals. Another study using deep learning methods showed that FDG-PET imaging can be used to predict AD an average of 75.8 months prior to its final diagnosis with 82% specificity and 100% sensitivity[19].

Taken together, our current findings and those of previous reports suggest that maintaining normal brain glucose metabolism is critical for cognitive resilience; therefore, therapeutic strategies for preventing or treating AD may need to shift focus from Aβ toward the preservation and restoration of normal brain metabolism. Interventions with this therapeutic strategy have been reported that use intranasal insulin administration and a ketogenic diet. Specifically, intranasal insulin therapy provides rapid delivery of insulin to the central nervous system via bulk flow along olfactory and trigeminal perivascular channels without adversely affecting blood insulin or glucose levels and has been shown to improve AD symptomology, although individual patient responses may depend on gender, APOE genotype and insulin formulation[57–59]. With regards to the potential benefits of a ketogenic diet, ketone bodies can function as an alternative fuel substrate in the brain when glucose is unavailable or when glucose metabolism is impaired due to insulin resistance[43,60–62]. One study showed that a ketogenic diet can modulate deposition of Aβ and Tau in the CSF of MCI patients in conjunction with its modulation of the gut microbiome and the production of short-chain fatty acids[63]. This finding is consistent with an animal study showing that a ketogenic diet enhanced Aβ clearance across the blood-brain barrier and improved the composition of the gut microbiome[64]. The gut microbiome produces secondary bile acids, and, as mentioned above, alterations of bile acid production have been observed in AD patients due to gut microbiome imbalances, suggesting another mechanism by which AD patients may benefit from therapeutic strategies aiming to restore normal brain metabolism like the ketogenic diet[47,48]. Another animal study showed that by modulating the gut microbiome with a prebiotic diet, mice with the human APOE ε4 gene had enhanced systemic metabolism and reduced neuroinflammatory gene expression, another hallmark of AD pathology[65]. Collectively, modulating metabolic function and the gut microbiome may have a profound impact on reducing the risk of AD.

Future efforts should include the continued collection of the A/T/N framework biomarkers to fill critical gaps in our understanding of how their expression is associated with AD and aging. In our model construction and analysis, we used CSF-pTau to fulfill the "T" component of the A/T/N framework[9]; however, imaging-derived biomarkers provide information about the location of the pathology in the brain that CSF-derived markers do not[8]. Therefore, future work is needed to incorporate Tau-PET imaging into the model[66]. In addition, glucose metabolism is tightly coupled with cerebral blood flow (CBF)[67,68], and neurovascular dysfunction also plays a critical role in cognitive impairment; thus, it will also be important to include CBF-MRI measures in the future for a more thorough representation of AD pathology. Indeed, Tau and CBF imaging data are currently available for only a small subset of the ADNI cohort, and thus it could not be incorporated into our model. Additionally, while the available dataset from ADNI has more male participants, it should be noted that AD disproportionately affects women[69]. Future efforts may be needed to re-evaluate the outcome when data from the female participants become more available.

In summary, we show that A/T/N biomarkers have cognitive impairment stage-dependent roles in AD, with Aβ and pTau better predicting LMCI and neurodegeneration (especially low glucose uptake) better predicting clinical AD. Our findings may partly explain the repeated failures of clinical trials attempting to treat AD by modifying the Aβ load: it may be too late to gain

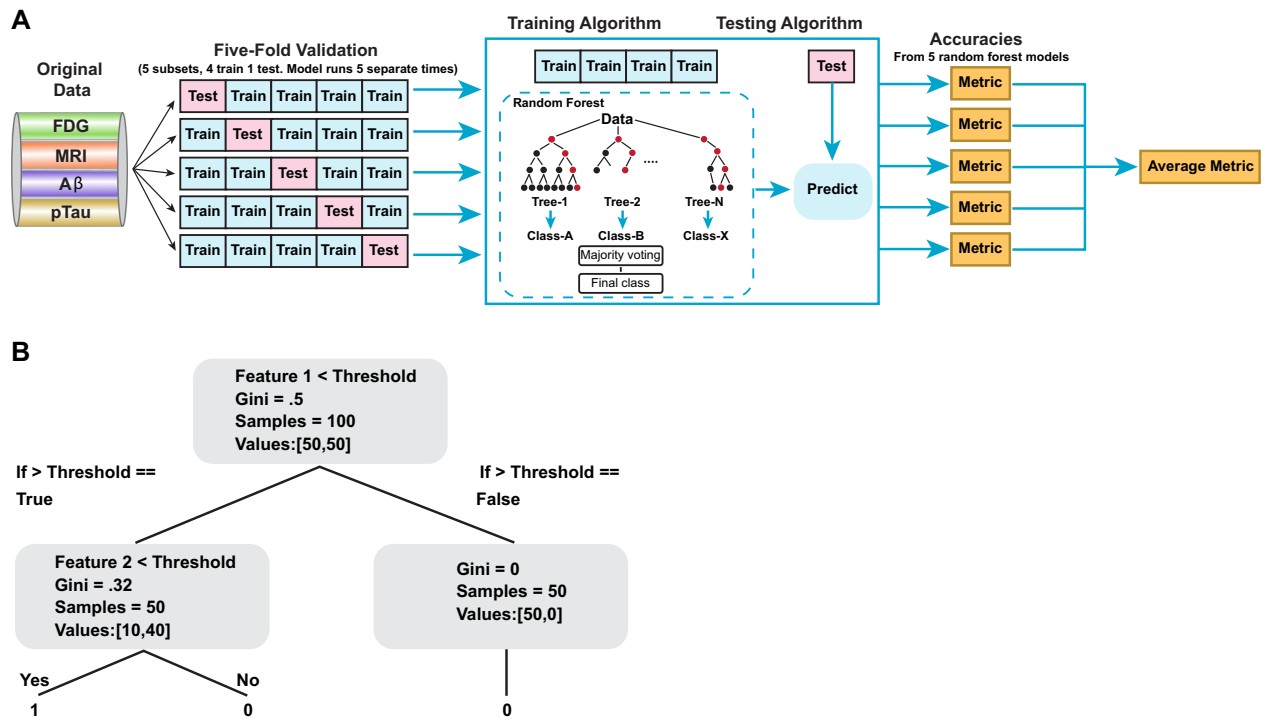

**Fig. 5 Flow chart of the random forest method used. a** Flow chart depicting the analysis used with the random forest method. The AD biomarkers from the original dataset were randomly split into five equal-sized subsets. For evaluation, each complete data copy was forwarded into a random forest (decision tree; see **b**) classifier model. Final predictions were calculated and features were ranked based on the prediction of the majority of trees within that training dataset. **b** Decision trees are classified in a binary fashion, where the split in the trees are from either true or false responses to feature thresholds based on Gini Impurity. "Purity" is a measure homogeneity, with "0" as maximal purity, and "1" as maximal impurity.

therapeutic benefit from the treatment of Aβ when patients have already progressed to the clinical AD stage. Therefore, our results imply that treatments for AD may also need to be disease stage-oriented: Aβ and tau may be appropriate targets early in the disease course, but the restoration of brain glucose metabolism should be explored as a treatment strategy for clinical AD. Our findings may influence the thinking in the field regarding AD progression and therapeutics.

## Methods

**Data pre-processing**. Study data were obtained from the ADNI database, a longitudinal multicenter study designed to develop clinical, imaging, genetic, and biochemical biomarkers for the early detection and tracking of Alzheimer's disease. Specifically, data were downloaded from the ADNI2 dataset within the ADNI database since these data contained all the biomarkers of interest for the present study. Specific details about the acquisition of the imaging measures have been reported elsewhere[70,71]. Briefly, all subjects were consented under the approval of the IRB at each testing site and scanned at 3 T for 3D T1-weighted volume, FLAIR, a long TE gradient echo volumetric acquisition for micro hemorrhage detection, arterial spin-labeling perfusion, resting state functional connectivity, and diffusion tensor imaging; all enrolled subjects were also scanned for [18F]fluorodeoxyglucose PET (FDG-PET) glucose uptake and [18F]florbetapir PET for amyloid imaging. The data were merged from five subset datasets within the ADNI2 dataset to achieve a final dataset for analysis consisting of demographic information, structural MRI volumes, FDG-PET SUVs, amyloid-PET SUVs, white matter hyperintensities, and CSF-ptau measurements. Age, gender, education, *APOE* ε4 carrier status, cognitive scores, and diagnosis and the structural MRI variables of ventricle volume, whole brain volume, entorhinal cortex volume, and hippocampal volume were extracted from the ADNIMERGE subset dataset. FDG-angular, FDG-temporal, and FDG-CingulumPost were extracted from the UC Berkeley FDG subset dataset. Aβ-frontal, Aβ-cingulate, Aβ-parietal, Aβ-temporal, Aβ-precuneus, and Aβ-hippocampus were extracted from the UC Berkeley AV45 subset dataset. Gray matter volume, white matter volume, and white matter hyperintensity were extracted from the UC Davis white matter hyperintensity volumes subset dataset. pTau concentration was extracted from the UPENN CSF Biomarkers Elecsys subset dataset. Missing values were imputed by selecting the twenty closest patients based on Euclidean distance with non-missing values in the same group and averaging these values. Most of the missing values appeared in the structural MRI data. Data imputation was performed on patients who had less than

three missing values. Patients with three or more missing values were deleted to avoid bias caused by excessive imputation.

**Machine learning analysis**. The random forest (RF) classification algorithm was used to assess the importance of all sixteen biomarker features in predicting the AD clinical diagnosis, as determined by the progression of cognitive impairment as a result of the disease process (CU, LMCI, or AD) (Table 2). The algorithm was chosen, as opposed to other traditional statistical (e.g., ANOVA) and machine learning methods, because (i) it is a robust classification method and (ii) it enables feature ranking. An RF is trained by fitting multiple decision trees, each to a different random subset of the examples and features of the full dataset. The predictions of these decision trees are then combined to yield a single consensus classification prediction. Given the trained RF, each feature is considered more important if decision trees constructed from subsets that include the feature give predictions that are more accurate. This is calculated by averaging the out-of-bag accuracy (i.e., the accuracy on examples there were not used when training the tree) of the individual decision trees that were trained using the corresponding feature.

We acknowledge that some of the features (e.g., brain volumetric measures) are impacted with age; therefore, we adjusted the feature values for age accordingly. Specifically, we used CU group dataset and performed a linear regression with the seven brain volumetric measures used in our model as the feature variables and age as the target regression variable. We applied the derived beta coefficients from the regression model to the brain volumetric measures of the whole dataset and trained these balanced brain volumetric measure values in our RF model. In the implementation, we used the function *sklearn.linear_model.LinearRegression* of the "scikit-learn" package to calculate the linear regression coefficients between brain volumetric measures and age.

Figure 5a illustrates the workflow of the feature ranking and accuracy performance using the random forest machine learning method. K-fold cross validation ($k = 5$) was used to evaluate the performance of the RF classification algorithm in predicting the AD clinical diagnosis. Using this strategy, the dataset was randomly partitioned into five equal parts, and five RF models were trained, each on a dataset consisting of four parts. Each of the trained RF models was evaluated based on the prediction performance on the corresponding omitted validation set. For evaluation, each complete data copy was forwarded into a random forest classifier model utilizing the Python scikit-learn library v0.21.3[72]. All default parameters were used for the *sklearn.ensemble.RandomForestClassifier* function, with the exception of the criterion parameter, where we used the *entropy* option. Specifically, decision trees are classified in a binary fashion where the split in the trees are from either true or false responses to feature thresholds. The

*RandomForestClassifier* decides the thresholds based on Gini Impurity. "Purity" is a measure as to how homogenous the samples are, with "0" as maximal purity, and "1" as maximal impurity. As the decision tree progresses down, the Gini values eventually decrease to 0 (Fig. 5b). Final predictions were calculated and features were ranked based on the prediction of the majority of trees within that training dataset. The resulting predictions were evaluated on their ability to correctly predict the AD clinical diagnosis in the validation dataset.

The cross-validated model prediction accuracy, receiver operating characteristic curve (ROC) and $F_1$ score were used to assess model performance.

The accuracy is calculated with the following Eq. (1):

$$\text{Accuracy} = \frac{\text{TP} + \text{TN}}{\text{TP} + \text{TN} + \text{FP} + \text{FN}}, \tag{1}$$

where TP is the true positives, TN the true negatives, FP the false positives, and FN the false negatives.

The $F_1$ score is calculated by the following Eq. (2):

$$F_1 = 2 \cdot \frac{precision \cdot recall}{precision + recall} \tag{2}$$

where precision is TP/(TP + FP) and recall is TP/(TP + FN). ROC curves compare the true positive rate and false positive rate at different decision thresholds and are often used to judge the performance of binary classifiers. $F_1$ scores combine precision and recall and are often used to evaluate models on imbalanced dataset, since it is possible to obtain high accuracy on imbalanced datasets simply by predicting the most common class. A high $F_1$ score indicates low false positives and low false negatives.

**Statistics and reproducibility**. In the Table 1, the overall dataset was initially evaluated for group differences in age, gender, education, *APOE* genotype, ethnicity, race, and cognitive test differences using non-parametric Kruskal–Wallis tests comparing the groups CU, LMCI, and AD using JMP 1.4 software. $\chi^2$-approximate values and *P*-values were documented to identify statistical significance. $\varepsilon^2$ values for effect sizes were calculated using the "rcompanion" package in R statistical software.

To verify the reproducibility of five-fold validation used in the RF analysis, we compared the results of accuracy and $F_1$ score from those using three-fold and ten-fold cross validations (Supplementary Table 1).

To verify the accuracy measurements validated using ROC, we also performed precision recall (PR) curves calculation (Supplementary Fig. 3). Precision-recall curve is another method to evaluate classification models, especially binary classification models where the dataset is imbalanced. The average of precision (AP) is calculated to determine the average precision score under different possible thresholds. We used the scikit learn package sklearn.model_selection. RandomizedSearchCV for hyperparameters optimization. The hyperparameters of the RF model is as follows: "n_estimators"=3600, "min_samples_split"=5, "min_samples_leaf"=8, "max_features"="auto", "max_depth"=50, and "bootstrap"=False.

We also used the SHapley Additive exPlanations (SHAP) technique to implement an additional feature ranking analysis. In our experiments, we applied the SHAP on Random Forest Classifier. Using the SHAP method as a reference for feature ranking analysis, the results showed similar feature importance ranking as RF (Supplementary Fig. 2).

Gradient tree boosting (GTB), another classification method from the *scikit-learn* package, was used as a comparison for the RF classification method (Supplementary Table 2). The same tree estimators from the RF method were used for GTB with all other default function parameters. The accuracies for the GTB method were similar to the RF method. The accuracy of the GTB classifiers were 72.30%, 71.26%, and 91.87%, respectively for CU vs. LMCI, LMCI vs. AD, and CU vs. AD clinical diagnosis. The model was also trained with 3- and 10-fold cross validation for comparison. There were minor difference in the feature rankings estimated using the GTB model as compared to the RF model but the same general patterns hold true: Aβ and pTau are important contributors to the prediction of early AD decline, but neurodegeneration, especially glucose hypometabolism, is a more important predictor of later AD decline.

**Pearson correlation analysis of cognitive performance**. Pearson correlation was used to evaluate linear relationships between individual biomarker features and cognitive function using the JMP 1.4 software (Figs. 2 and 3). Pearson correlation coefficient is calculated by the covariance of two variables over the product of their standard deviation. The value range of Pearson correlation coefficient is from −1 to 1 with a higher absolute value indicating a stronger association and the sign indicating a positive or negative association between the two variables.

**Calculation of biomarker values**. In the Table 5, biomarker values were calculated for the different diagnosis groups and compared using two-sided Wilcoxon rank-sum tests. Z-score test statistics were calculated using JMP 1.4 software and effect sizes *r* were calculated with *r* (=$Z/(\sqrt{N_{obs}})$). Amyloid standard uptake values (SUVs) were intensity normalized to the whole cerebellum and volume was normalized by dividing by the region of interest (ROI) in cubic centimeters ($cm^3$). FDG SUVs were normalized according to metaROIs described elsewhere[73]. Briefly, a set of pre-defined regions of interest (FDG-ROIs) were developed by identifying regions cited frequently in FDG-PET studies of AD and MCI patients. All coordinates of significant voxels were transformed into MNI space. Intensity values were generated for coordinates that reflected a combination of the *Z*-scores associated with the coordinate. The volumes were intensity normalized using the maximum value, and volume was normalized by dividing by the ROI in cubic centimeters ($cm^3$).

**Reporting summary**. Further information on research design is available in the Nature Research Reporting Summary linked to this article.

## Data availability
The datasets analyzed during the current study are available in the Alzheimer's Disease Neuroimaging Initiative (ADNI) repository, http://adni.loni.usc.edu/.

## Code availability
The code used in analysis for the current study are available in a GitHub control repository, https://github.com/linbrainlab/machinelearning.git.

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

## Acknowledgements

Data used in preparation of this article were obtained from the Alzheimer's Disease Neuroimaging Initiative (ADNI) database (adni.loni.usc.edu). As such, the investigators within the ADNI contributed to the design and implementation of ADNI and/or provided data but did not participate in analysis or writing of this report. A complete listing of ADNI

investigators can be found at: http://adni.loni.usc.edu/wp-content/uploads/how_to_apply/ADNI_Acknowledgement_List.pdf Data collection and sharing for this project was funded by the Alzheimer's Disease Neuroimaging Initiative (ADNI) (National Institutes of Health Grant U01 AG024904) and DOD ADNI (Department of Defense award number W81XWH-12-2-0012). ADNI is funded by the National Institute on Aging, the National Institute of Biomedical Imaging and Bioengineering, and through generous contributions from the following: AbbVie, Alzheimer's Association; Alzheimer's Drug Discovery Foundation; Araclon Biotech; BioClinica, Inc.; Biogen; Bristol-Myers Squibb Company; CereSpir, Inc.; Cogstate; Eisai Inc.; Elan Pharmaceuticals, Inc.; Eli Lilly and Company; EuroImmun; F. Hoffmann-La Roche Ltd and its affiliated company Genentech, Inc.; Fujirebio; GE Healthcare; IXICO Ltd.; Janssen Alzheimer Immunotherapy Research and Development, LLC.; Johnson and Johnson Pharmaceutical Research and Development LLC.; Lumosity; Lundbeck; Merck and Co., Inc.; Meso Scale Diagnostics, LLC.; NeuroRx Research; Neurotrack Technologies; Novartis Pharmaceuticals Corporation; Pfizer Inc.; Piramal Imaging; Servier; Takeda Pharmaceutical Company; and Transition Therapeutics. The Canadian Institutes of Health Research is providing funds to support ADNI clinical sites in Canada. Private sector contributions are facilitated by the Foundation for the National Institutes of Health (www.fnih.org). The grantee organization is the Northern California Institute for Research and Education, and the study is coordinated by the Alzheimer's Therapeutic Research Institute at the University of Southern California. ADNI data are disseminated by the Laboratory for Neuro Imaging at the University of Southern California. We also thank Dr. Erin Abner of University of the Kentucky for statistical support and Mr. Thomas Dolan of the University of Kentucky for graphical design. This research was supported by NIH grants R01AG054459 and RF1AG062480 to A.-L.L., R03AG063250, R03AG054936, and R01 LM012535 to K.N., ADNI Psychometrics R01AG029672 to P.K.C., training grant T32DK007778 to L.M.Y., and NSF CAREER award (IIS #1553116) to N.J. This study is also a part of the collaboration from the Friday Harbor Psychometrics Conference sponsored by R13AG030995 (PI: Mungas).

## Author contributions

T.C.H., X.X., D.M., K.N., P.K.C., F.E., D.A.Z., Q.C., and A.-L.L. participated in study design. T.C.H., X.X., C.W., D.M., K.N., and A.-L.L. participated in ADNI data identification and consolidation. X.X., C.W., D.M., G.L., Q.C., and N.J. participated in machine learning analyses. T.C.H., X.X., and A.-L.L. participated in Pearson correlation and statistical analyses. K.N., P.K.C., F.E., D.A.Z., and A.-L.L. participated in data verification and result interpretation. T.C.H., X.X., C.W., D.M., K.N., P.K.C., F.E., D.A.Z., G.L., Q.C., L.M.Y., N.J., and A.-L.L. participated in manuscript writing.

## Competing interests

The authors declare no competing interests.
