## [Peer Review File · Communications Biology]

Reviewers' comments:

Reviewer #1 (Remarks to the Author):

The issue of stage dependent biomarkers is an important topic to both understand progression of AD and to the development of therapeutic strategies. Overall, this is a well written report with intriguing findings.

Fundamentally, this analysis parses out the relative contribution of AD pathologies and assigns these variables to a stage based on cognitive performance.

Overall the outcomes of the AI / machine learning approach conducted by Lin and colleagues matches very well with those reported earlier by Jack and colleagues: Update on hypothetical model of Alzheimer's disease biomarkers *Lancet Neurol.* 2013 Feb; 12(2): 207–216.

Since the findings by Lin and colleagues replicate those in earlier clinical analyses, the real strength of this study is the ability to generate algorithms and correlations for computational analyses that are consistent with clinical, pathological and statistical assessments. The challenge is to translate analysis of data that is readily available collected and aggregated over the course of decades and to the capacity to individual patient assessment in real time.

Reviewer #3 (Remarks to the Author):

In this study a Random Forest model was constructed to differentiate Alzheimer's disease, late mild cognitive impairment and cognitively unimpaired patients using PET, MRI and CSF biomarkers. The model was then used to interpret the contribution of these factors with feature importance analysis and by examining patterns of correlation with several cognitive scores.

Major comments

(1) Although it is interesting to see a machine learning method being used primarily for the interpretation of variable effects, the main question for most readers will be why this approach was chosen rather than a more common statistical method, like multivariate ANOVA? The trade-offs and rationale for this choice should be explained in detail in the introduction.

(2) In table 1 authors have shown that several confounding variables may potentially impact the results. However, no step appears to have been taken to address this in subsequent analysis. In particular, brain volumetric measures are known to change with age and therefore could be a proxy for this confounding feature. To ensure good quality results, this should be dealt with, e.g. by using covariate balancing.

(3) Feature importance is now largely superseded by a more advanced and accurate SHAP method (Lundberg et al., 2017). The implementation is readily compatible with the described procedure and adding this type of analysis to the paper will greatly improve the interpretability of the results.

(4) To make the most out of the modern machine learning methods it is particularly importance to perform hyperparameter optimization. As this was not mentioned anywhere, I am assuming it was not done in this case. Either way, thorough evaluation should be done to show the effect of at least key RF parameters on the performance of the model and the results reported in the paper.

Minor comments

(1) Re: "...random forest machine learning method because it not only has high prediction accuracy due to its use of multiple decision trees". High accuracy is a property of a particular model, not an algorithm and even very simple algorithms can be used to create very accurate models in specific cases - consider rephrasing.

(2) Some additional information should be added to fully characterize stability and quality of the constructed model: performance at other K-folds (e.g. 3, 10) and PR curves in addition to ROC curves.

(3) As noted by the authors, some of the features used in the model are correlated, and this should be characterized further - e.g. by including some plots/heatmaps to show the correlation structure with respect to the overall vs. selected features.

(4) The discussion emphasizes that reported findings are quite consistent with several previous studies. As Communications Biology prioritizes novelty, if there are any specific novel insights from this study, they should be identified more explicitly.

(5) Panels in Fig. 2 should be in the same order to make it easier for the reader.

We thank the reviewers for the thoughtful and constructive comments. We are pleased that we were able to address all the concerns and made the changes accordingly. In particular, we significantly revised the Methods and Results sections to accommodate the reviewers' suggestions and added five new figures/tables as the Supplementary materials. We also revised Discussion accordingly. The changes are highlighted in yellow in the manuscript. The point-by-point response can be found below.

Reviewer #1

Overall the outcomes of the AI / machine learning approach conducted by Lin and colleagues matches very well with those reported earlier by Jack and colleagues: Jack, Knopman et al. Lancet Neurol. 2013 Feb; 12(2): 207–216.

Response: We thank the reviewer for the suggestion. We have included this paper in Discussion added it as a reference (Page 14, Lines 304-306).

Since the findings by Lin and colleagues replicate those in earlier clinical analyses, the real strength of this study is the ability to generate algorithms and correlations for computational analyses that are consistent with clinical, pathological and statistical assessments. The challenge is to translate analysis of data that is readily available collected and aggregated over the course of decades and to the capacity to individual patient assessment in real time.

Response: We agree. We have included this information in our Discussion (Page 14, Lines 306-310).

Reviewer #2

Major Comments:

1) Results (page 9, line 222): While the paper describes the correlation of the various AD biomarkers with broad measures such as composite scores of memory and executive functioning, there is no description in the paper of the correlations between the individual measures themselves. This information would be helpful to the reader in showing which measures are uniquely informative versus which measures are highly correlated and thus likely to show similar levels of feature importance in random forest analysis.

Response: We have included a heatmap to illustrate how the features correlate with each other (Supplementary Figure 1; see below) and have changed the Results accordingly (Page 6, Lines 147-150).

Supplementary Figure 1. Correlation Heatmap of biomarker features. Heatmap depicting how the biomarker features were correlated with one another. The color legend is representative of Pearson correlation r values.

2) Discussion (page 16, line 368): In the description of potential limitations/concerns, it should be noted that the distribution of all three groups are predominantly male, whereas Alzheimer’s disease disproportionately affects women (e.g., most study case groups are 55-65% female). It should be discussed whether this is a feature of study design or completeness of data, as it may have some effect on the inferences that can be drawn.

Response: We thank the reviewer to point this out. We have expanded our discussion to reference this potential limitation (Pages 18-19, Lines 438-441) as follows.

“Additionally, while the available dataset from ADNI has more male participants, it should be noted that AD disproportionately affects women⁶⁹. Future efforts may be needed to re-evaluate the outcome when data from the female participants become more available”.

Minor Comments:

1) General: Wherever the gene APOE is referenced, it should be italicized, and all instances referring to the APOE $\epsilon 4$ allele (including in tables) should be printed as “APOE $\epsilon 4$ ” and not as “APOE4.”

Response: We have changed the formatting accordingly throughout the manuscript.

2) Discussion (page 17, line 397): The controversial term “type 3 diabetes” is often used to indicate that aberrant glucose metabolism in the brain mirrors the effects of insulin resistance issues elsewhere in the body, however this characterization is prone to oversimplification and misunderstanding. A more pragmatic statement might be to say that the metabolic abnormalities present in AD are often likened to a form of diabetes of the brain, while keeping the same references.

Response: We have made the suggested change on Page 17, Lines 396-397.

3) Methods (page 19, line 452): Additional citations for papers describing the collection of ADNI MRI and glucose metabolism variables should be included, as details on the collections of the phenotypes will be important for some readers. Those details do not need to be reproduced in the manuscript, but links or citations for that information should be provided.

Response: We have included the suggested references (Page 19, lines 459-460).

Reviewer #3

Major comments

(1) Although it is interesting to see a machine learning method being used primarily for the interpretation of variable effects, the main question for most readers will be why this approach was chosen rather than a more common statistical method, like multivariate ANOVA? The trade-offs and rationale for this choice should be explained in detail in the introduction.

Response: We appreciate the reviewer for the thoughtful concern. As the major goal of the study is being able to rank the features as well as determine the accuracy of the prediction, we chose to use Random Forest method, a machine learning algorithm, as it fits for the purpose. Other more common statistical methods, including ANOVA, would not be able provide the information requested by the study. We included this comment on Page 20, Line 482.

(2) In table 1 authors have shown that several confounding variables may potentially impact the results. However, no step appears to have been taken to address this in subsequent analysis. In particular, brain volumetric measures are known to change with age and therefore could be a proxy for this confounding feature. To ensure good quality results, this should be dealt with, e.g. by using covariate balancing.

Response: We thank the reviewer for the comment. We adjusted for age in the revision. Specifically, we used the linear regression model between the 6 brain volumetric measures and age on the CU group, then applied the fitted coefficients across all other groups to achieve the covariate balancing before running RF. In the implementation, we used the function *sklearn.linear_model.LinearRegression* of scikit-learn package to calculate the linear regression coefficients between brain volumetric measures and age. The changes can be seen on Page 5, Lines 134-135 and Page 20, Lines 491-498. We also updated the results in Table 3 accordingly.

(3) Feature importance is now largely superseded by a more advanced and accurate SHAP method (Lundberg et al., 2017). The implementation is readily compatible with the described procedure and adding this type of analysis to the paper will greatly improve the interpretability of the results.

Response: We thank the reviewer for the suggestion. In this revision, we used the SHapley Additive exPlanations (SHAP) technique to implement an additional feature ranking analysis. We found that the results from SHAP analyses were consistent with those from Random Forest. We made the changes accordingly in the Methods and Results (Page 7, Lines 181-183 and Page 23, Lines 566-569) and added Supplementary Figure 2.

CU vs. LMCI:

LMCI vs. AD:

CU vs. AD:

Supplementary Figure 2. SHAP Analysis depicting biomarker feature ranking importance. Comparison of feature ranking analysis from implementation of the SHapley Additive exPlanations (SHAP) technique. The bar plots shows feature impacts on the cognitively unimpaired (CU) vs late mild cognitive impairment (LMCI) analysis, the LMCI vs Alzheimer’s disease (AD) analysis, and the CU vs AD analysis.

(4) To make the most out of the modern machine learning methods it is particularly importance to perform hyperparameter optimization. As this was not mentioned anywhere, I am assuming it was not done in this case. Either way, thorough evaluation should be done to show the effect of at least key RF parameters on the performance of the model and the results reported in the paper.

Response: We agree. We added hyperparameter optimization in this revision. We changed the Methods accordingly (Pages 22-23, Lines 561-564), and as follows.

“We used the scikit learn package `sklearn.model_selection.RandomizedSearchCV` for hyperparameters optimization. The hyperparameters of the RF model is as follows: `'n_estimators'=3600, 'min_samples_split'=5, 'min_samples_leaf'=8, 'max_features'='auto', 'max_depth'=50, and 'bootstrap'=False`”.

Minor comments

(1) Re: “...random forest machine learning method because it not only has high prediction accuracy due to its use of multiple decision trees”. High accuracy is a property of a particular model, not an algorithm and even very simple algorithms can be used to create very accurate models in specific cases - consider rephrasing.

Response: We thank the reviewer for the suggestion. We have rephrased the sentence accordingly (Page 4, Line 93).

(2) Some additional information should be added to fully characterize stability and quality of the constructed model: performance at other K-folds (e.g. 3, 10) and PR curves in addition to ROC curves.

Response: We have added Precision Recall (PR) curves to evaluate our Random Forest models (Supplementary Figure 3) and have changed the Methods and Results accordingly (Page 9, Line 205 and Pages 22-23, Lines 557-564). We also added the results with 3 and 10 K-folds calculations in addition to the original 5 K-fold outcomes (Supplementary Table 1) and have changed the Methods and Results accordingly (Page 9, Lines 202-203, and Page 22, Line 555).

Supplementary Table 1. Accuracy results of all features and Top 8 features in RF model from A) three- and B) ten-fold cross validation

A) K-fold=3

All Features			
	CU vs. LMCI	LMCI vs. AD	CU vs. AD
Accuracy (%)	72.26	72.42	90.90
F1-score (%)	72.15	72.37	90.88
Top 8 Features			
	CU vs. LMCI	LMCI vs. AD	CU vs. AD
Accuracy (%)	72.34	71.60	91.40
F1-score (%)	72.21	71.59	91.38

B) K-fold=10

All Features			
All Features	CU vs. LMCI	LMCI vs. AD	CU vs. AD
Acc.	72.15	71.45	90.62
F1-score	71.92	71.03	90.60
Top 8 Features			
Top 8 Features	CU vs. LMCI	LMCI vs. AD	CU vs. AD
Acc.	71.19	70.49	91.86
F1-score	70.98	70.15	91.81

Precision recall (PR) curves

Supplementary Figure 3. Comparison of precision recall (PR) curves between all 16 biomarker features (top) and the top 8 biomarker features (bottom) from the three diagnosis participant group comparisons: cognitively unimpaired (CU) vs late mild cognitive impairment (LMCI), LMCI vs Alzheimer’s disease (AD), and CU vs AD. The Average of Precision (AP) is calculated to see the average precision of each model under different possible thresholds.

3) As noted by the authors, some of the features used in the model are correlated, and this should be characterized further – e.g. by including some plots/heatmaps to show the correlation structure with respect to the overall vs. selected features.

Response: We have included a heatmap to illustrate how the features correlate with each other (Supplementary Figure 1) and have changed the Results accordingly (Page 6, Lines 147-150).

Supplementary Figure 1. Correlation Heatmap of biomarker features. Heatmap depicting how the biomarker features were correlated with one another. The color legend is representative of Pearson correlation r values.

(4) The discussion emphasizes that reported findings are quite consistent with several previous studies. As Communications Biology prioritizes novelty, if there are any specific novel insights from this study, they should be identified more explicitly.

Response: We have made efforts to identify the novelty more explicitly (Page 14, Lines 304-310).

(5) Panels in Fig. 2 should be in the same order to make it easier for the reader.

Response: The arrangement of the panels in Figure 2 is according to the order of the correlation coefficient (r value, from the highest to the lowest); therefore, the order would be different in each comparing condition. We made this clearer on Page 10, Lines 231-232 and in the figure legends of Figures 2 and 3.

REVIEWERS' COMMENTS:

Reviewer #2 (Remarks to the Author):

Review for Nature Communications Biology of the revised manuscript:

β -Amyloid and tau drive early Alzheimer's disease decline while glucose hypometabolism drives late decline

T. Hammond, X. Xing, D. Ma, K. Nho, P.K. Crane, F. Elahi, D. Ziegler, G. Liang, Q. Cheng, N. Jacobs, A.-L. Lin, Alzheimer's Disease Neuroimaging Initiative (ADNI)

Overview: This manuscript provides an insightful look into biomarkers categorized by the "A/T/N" framework (comprising A β [beta-amyloid] biomarkers, Tau biomarkers, and biomarkers of neurodegeneration or neuronal injury) and their predictive utility in identifying different stages in Alzheimer's disease (AD) progression using random forest machine learning on participant data from the Alzheimer's Disease Neuroimaging Initiative (ADNI) dataset. The analysis utilizes this approach to identify biomarkers correlations of relevance, and is able to discern that markers of A β and Tau (A/T biomarkers) are more predictive of pre-clinical cognitive decline while markers of glucose hypometabolism (N biomarkers) are predictive of later dementia status. This study adds further validity to a paradigm that is being used more widely to characterize dementia progression.

All major and minor concerns have been well-addressed by the authors, and only a few grammatical and typographical suggestions are suggested, as noted in minor comments. With these minor changes (that do not require additional review), this study is recommended for publication.

Minor Comments:

- 1) Though the acronym "PET" is defined as "positron emission tomography" in Table 2, it should probably be defined the first time it appears in the text (page 4, line 90).
- 2) Likewise, though the acronym "FDG" is elaborated as "fluorodeoxyglucose" in a footnote to Table 3, it should probably be defined the first time it appears in the text (page 4, line 91).
- 3) This is a minor stylistic suggestion, but the authors' response to the second comment of Reviewer #1 was to integrate their suggested conclusion verbatim into the text. It is suggested that the authors either (a) confirm it is okay with the reviewer to use their phrase verbatim without attribution, or (b) rephrase or paraphrase the statement at least a little to avoid any potential concerns. It's likely not an issue at all, and may not even be regarded as necessary, but it's a safe step to avoid accusations of plagiarism or misattribution, and it should be a relatively easy fix. When the borrowing of wording is for minor issues like phrasing or makes up only part of a sentence, it's not a big concern, but two full sentences of a new conclusion might be viewed differently.
- 4) Minor typographical/grammatical suggestion: Add single quote to ` and include the word "the" in this line on page 20, line 497: "... sklearn.linear_model.LinearRegression of the `scikit-learn` package..."

Reviewer #3 (Remarks to the Author):

All of my previous concerns have been adequately addressed in this version of the manuscript - I am happy to recommend this paper for publication.

We thank the Reviewer 2 for the additional and thoughtful comments. We are pleased that we were able to address all the concerns and to adjust the manuscript to comply with Nature Communications Biology format requirements. The responses to the reviewer are highlighted in yellow and those to the Editor are highlighted in turquoise in the manuscript. The point-by-point response can be found below.

Reviewer #2

Though the acronym “PET” is defined as “positron emission tomography” in Table 2, it should probably be defined the first time it appears in the text (page 4, line 90).

Response: We thank the reviewer for the suggestion. We have defined PET at “positron emission tomography” on page 4, line 93.

Likewise, though the acronym “FDG” is elaborated as “fluorodeoxyglucose” in a footnote to Table 3, it should probably be defined the first time it appears in the text (page 4, line 91).

Response: We agree. We have define FDG as “fluorodeoxyglucose” on page 4, line 94.

This is a minor stylistic suggestion, but the authors’ response to the second comment of Reviewer #1 was to integrate their suggested conclusion verbatim into the text. It is suggested that the authors either (a) confirm it is okay with the reviewer to use their phrase verbatim without attribution, or (b) rephrase or paraphrase the statement at least a little to avoid any potential concerns. It’s likely not an issue at all, and may not even be regarded as necessary, but it’s a safe step to avoid accusations of plagiarism or misattribution, and it should be a relatively easy fix. When the borrowing of wording is for minor issues like phrasing or makes up only part of a sentence, it’s not a big concern, but two full sentences of a new conclusion might be viewed differently.

Response: We thank the reviewer for the suggestion. We have rephrased the statements to avoid any potential concerns (Page 8, Line 250-253).

Minor typographical/grammatical suggestion: Add single quote to “ and include the word “the” in this line on page 20, line 497: “... sklearn.linear_model.LinearRegression of the ‘scikit-learn’ package...”

Response: We agree. We have followed the suggestion and altered the text on Page 14, Line 441.